# Congenital Lung Malformations: Clinical and Functional Respiratory Outcomes after Surgery

**DOI:** 10.3390/children9121881

**Published:** 2022-12-01

**Authors:** Andrea Farolfi, Michele Ghezzi, Valeria Calcaterra, Giovanna Riccipetitoni, Gloria Pelizzo, Sara Costanzo, Emma Longoni, Annalisa De Silvestri, Nicolò Garancini, Salvatore Zirpoli, Gianvincenzo Zuccotti

**Affiliations:** 1Department of Pediatrics, Vittore Buzzi Children’s Hospital, 20154 Milano, Italy; 2Pediatric and Adolescent Unit, Department of Internal Medicine and Therapeutics, University of Pavia, 27100 Pavia, Italy; 3Pediatric Surgery Unit, Fondazione IRCCS Policlinico San Matteo, University of Pavia, 27100 Pavia, Italy; 4Department of Biomedical and Clinical Science “L. Sacco”, University of Milano, 20154 Milano, Italy; 5Pediatric Surgery Department, Vittore Buzzi Children’s Hospital, 20154 Milano, Italy; 6Biometry & Clinical Epidemiology, Scientific Direction, Fondazione IRCCS Policlinico San Matteo, 27100 Pavia, Italy; 7Radiology Department, Vittore Buzzi Children’s Hospital, 20154 Milano, Italy

**Keywords:** congenital lung malformations, pediatric surgery, children, pulmonary function test

## Abstract

Congenital lung malformations (CLMs) involve anomalies of the lungs and respiratory tree such as congenital pulmonary airway malformation (CPAM), pulmonary sequestration (PS), bronchogenic cysts, congenital lobar emphysema, and bronchial atresia (BA). Although symptomatic lesions require surgical resection, the appropriateness of surgery for patients with asymptomatic malformations is a matter of ongoing debate. Limited data are available concerning the long-term follow-up of affected subjects. In this study, we sought to evaluate the long-term clinical and functional respiratory outcomes in children with CLMs who underwent surgical resection. We carried out a retrospective analysis of 77 children with CLMs who underwent pulmonary resection with at least one year of follow-up. The most common diagnoses were CPAM (50.65%), hybrid lesions (25.97%), lobar emphysema (11.69%), and PS (5.19%). The most common surgical approaches were lobectomy (61.3%), segmentectomy (10.7%), and pneumonectomy (5.3%). Acute post-surgery complications occurred in 31.2% of children. In addition, 73.7% experienced long-term complications, and we found no correlation between the presence of these complications and the sex of the patients, their age at time of surgery, the type of surgery undergone, the presence of symptoms prior to intervention, or acute complications after surgery. Pulmonary function tests revealed FEV1 Z-scores of <−2 SDs in 16 patients, and we found a significant correlation between pneumonectomy and the development of lung function deficit (*p* = 0.031). In conclusion, clinical and functional respiratory complications may occur in children with CLMs who undergo surgical resection. Long-term monitoring is needed to improve the management of asymptomatic patients.

## 1. Introduction

Congenital lung malformations (CLMs) represent 5–18% of all congenital abnormalities, with a cumulative incidence of 30–42 cases per 100,000 subjects [1]. CLMs include different conditions such as congenital pulmonary airway malformation (CPAM), pulmonary sequestration (PS), bronchogenic cysts, congenital lobar emphysema, and bronchial atresia (BA). These represent a spectrum of developmental anomalies pathogenetically linked to the obstruction of the developing fetal airway [2]. Diagnosis of a particular condition is made after birth using thoracic MRI or CT scans.

Occasionally, CLMs cause mediastinal compression or shift, with hydrops and fetal death, and the subsequent need for intrauterine interventions [3]. Most lesions decrease in size during late pregnancy, and most affected children (>75%) are asymptomatic at birth [4].

There is scientific consensus regarding the need for the surgical resection of symptomatic lesions; however, in cases of clinically silent malformations, opinion is divided between surgical and more conservative approaches. Though this debate is ongoing, it has been little addressed in the published literature to date [5].

Potential postnatal complications of CLMs include infections, hemothorax, air embolism, and pneumothorax. Complications occur in about 3.2% of non-operative patients [6]. The relationship between CLMs and malignancy is another matter of debate. Even the most sophisticated imaging techniques have a limited ability to predict the final pathological diagnosis and to differentiate benign forms from those currently or potentially malignant, e.g., pleuropulmonary blastoma, because of the high frequency of mixed forms and the overlap of histopathologic elements among the entities. Moreover, among CLM cases, the absolute number of malignant neoplasia cases is very low. Advocates of a conservative treatment approach claim that the number of children who must be treated with elective resection to prevent one neoplasm is simply too high to justify the intervention [7]. However, critics of the conservative approach to the management of asymptomatic patients highlight issues such as the effects of radiation exposure due to the repeated CT scans needed for monitoring the evolution of CLMs, the risk of complications, and patients lost to follow-up [8].

At present, only limited long-term follow-up data are available for CLM patients. However, some recent retrospective studies have considered symptomatic and asymptomatic patients, operated upon at different ages, with different surgical approaches and with differing experiences of symptoms after intervention, which one study found in 40% of children [9,10,11,12].

Today, lobectomy is the most common surgical approach [5], and it is generally well-tolerated [5,13]. The long-term quality of respiratory function following surgical resection is very much dependent on the amount of lung tissue that is resected, but the timing of resection may also be significant [10,11]. Some evidence suggests that early surgery (<6 months of age) may allow for an easier operative intervention, a shorter recovery time, and possible compensatory lung growth, with potentially fewer complications [14].

In this study, we sought to evaluate the long-term clinical and functional respiratory outcomes in children with CLMs who underwent surgical resection and, thus, provide more information to assist with the management of lesions in the future.

## 2. Patients and Methods

### 2.1. Patients

We conducted a retrospective analysis of 77 children, with diagnoses of CLMs, referred to the Pediatric Pneumology outpatient clinics of Vittore Buzzi Children’s Hospital, Milan, Italy. All had undergone pulmonary resection between 2000 and 2020 and had at least 1 year of pulmonary follow-up.

We reviewed medical records to determine patient demographics, clinical features, the type of surgery undergone, imaging diagnoses, and histopathologic reports. We also reviewed early and long-term complications.

### 2.2. Methods

#### 2.2.1. Classification of the Complications

We classified clinical complications as early if they occurred during hospitalization for surgery or in the following two weeks. Possible early complications included pneumothorax, hemorrhage, anesthetic complications, and infection.

We classified adverse events occurring more than one month after surgery as late complications. We carried out a retrospective analysis of data from clinical records and determined significant long-term complications to be as follows: (1) at least one episode of cough continuing for more than two months every year [15]; (2) more than three lower-airway respiratory infections diagnosed by a doctor every year; (3) wheezing requiring medical therapy with inhaled bronchodilators and/or antibiotics; (4) reduced exercise tolerance; (5) chest deformity and scoliosis.

#### 2.2.2. Respiratory Function Tests

We recorded long-term respiratory outcomes by analyzing the results of respiratory function tests (PFT) carried out by global spirometry and/or body plethysmography methods.

Dynamic lung volumes were measured using spirometry, performed according to American Thoracic Society guidelines [16]. We documented the forced expiratory volume in 1 s (FEV1) and forced vital capacity (FVC) and then calculated FEV1/FVC. We calculated standard deviation scores (SDs) according to the Global Lung Initiative 2012, with −1.64 to +1.64 SDS considered as normal range [17].

Static lung volumes, measured by body plethysmography, when available, were collected and expressed in terms of the residual volume (RV), total lung capacity (TLC), RV/TLC, and airway resistance (RAW). We calculated SDs for static lung volumes and diffusion capacity according to the Utrecht data set [18].

### 2.3. Statistical Analysis

All quantitative data were summarized as mean and standard deviation (SD) or median and IQR (range), as appropriate. We used the Shapiro–Wilk method to assess the normality of data. Qualitative variables were expressed as counts and percentages. The association of categorical variables was assessed with chi-squared or Fisher’s exact test. The statistical significance of the continuous variable comparisons was assessed using unpaired Student’s *t*-test. To determine possible relationships between respiratory function tests, we carried out a correlational analysis (Pearson or Spearman, according to the achieved normality). A *p*-value below 0.05 was considered statistically significant. All data analysis was performed with the STATA statistical package (release 15.1, 2017, Stata Corporation, College Station, TX, USA).

## 3. Results

### 3.1. Demographics and Baseline Data

We analyzed the clinical records of 77 children (38 females and 39 males), with a median age at first evaluation of five months (IQR 7–36 months), who were diagnosed with CLM and then treated surgically.

Initial diagnostic suspicion was the result of fetal ultrasound in 40 patients, which was then confirmed by a fetal MRI in 19 cases (47.5%). Among these 19 patients, the prenatal diagnosis was PS in 3 cases, CPAM in 9 cases, congenital lobar emphysema in 1 case, and hybrid lesions in 6 cases. CLM was postnatally diagnosed by a thorax CT or thorax MRI in 19 out of 77 cases (24.7%).

The most frequent diagnosis was CPAM (50.65%), followed by hybrid lesions (25.97%), lobar emphysema (11.69%), and PS (5.19%). None of our patients presented with BA, and other types of malformations constituted 6.49% of the total study population. Hybrid lesions involved combinations of CPAM with PS, CPAM with BA, or CPAM with congenital lobar emphysema.

In 73 cases, histological examination data was available in the form of official reports. These confirmed the pre-resection diagnostic hypothesis in 36 of the 73 patients (49.32%), though without strong agreement (kappa = 0.32, *p* = <0.01).

Initial CPAM diagnoses were later histologically classified as hybrid lesions in eight cases, as lobar emphysema in four cases, as PS in one case, as bronchogenic cysts in one case, and as other malformations in two cases. Similarly, two cases involving an initial diagnosis of congenital lobar emphysema were later reclassified histologically: one as BA and another as hybrid lesions. Finally, two patients originally diagnosed with other malformations were later re-classified histologically: one as CPAM and the other as hybrid lesions. Histological re-classification especially affected patients initially diagnosed with hybrid lesions; of these, 10 were re-diagnosed with PS, 3 with CPAM, 1 with congenital lobar emphysema, 2 with BA, and 1 with another type of malformation.

### 3.2. Surgical Treatment

All 77 patients were treated surgically, and we recorded the presence or absence of symptoms before surgery in all cases. Twenty-eight patients had symptoms before surgery (36.4%); of these, seventeen experienced dyspnea (22.1%), six experienced cough (7.8%), seven experienced pneumothorax (9%), and seven experienced recurrent pulmonary infections (9%). Forty-nine patients were asymptomatic before surgery.

Surgical procedure documentation was only available in 75 cases, so we considered only these for study purposes.

The most frequent type of resection was lobectomy. Forty-six of the seventy-five patients underwent this procedure (61.3%), while eight patients underwent segmentectomy (10.7%) and four underwent pneumonectomy (4/75, 5.3%). Seventeen of the seventy-five (22.7%) patients underwent wedge resections. Thoracotomy was performed in 67.8% of patients; the remaining 32.2% underwent thoracoscopy.

Twenty-four patients out of seventy-seven (31.2%) experienced acute post-surgery complications. Descriptions of acute post-operative complications are set out in Table 1.

The median age at intervention of the patients with acute complications was 3.5 months (IQR 0.5–10). For the patients without post-surgery complications, the median age at intervention was 5 months (IQR 1–11.2).

The presence of these complications was not statistically related to patient sex, the timing of the surgery (before or after 6 months of life), the surgical method involved (lobectomy, segmentectomy, or pneumonectomy), or the presence of symptoms prior to intervention (*p* > 0.05). In particular, we found no correlation between the development of early post-surgical complications and the choice of a specific surgical approach (lobectomy, segmentectomy, or pneumonectomy) (*p* = 0.16) or the choice of thoracotomy or thoracoscopy (*p* = 0.657).

### 3.3. Long-Term Follow-Up

We recorded long-term adverse events for all patients, except one who was lost to follow-up. Fifty-six out of these seventy-six patients (73.7%) experienced long-term complications, such as chronic cough, recurrent lower-airway infections, wheezing, poor tolerance to exercise, or orthopedic complications. (Table 2).

The development of chest or column deformities was not statistically associated with patient sex, type of surgery, the presence of symptoms prior to surgery, or short-term complications after the intervention (all *p* > 0.05).

Long-term complications were not statistically related to age at time of surgery, patient sex, presence of symptoms pre-intervention, type of surgery, or the presence of short-term complications after surgery (all *p* > 0.05).

The choice of a specific surgical approach (lobectomy, segmentectomy, or pneumonectomy) was not correlated with the development of thorax abnormalities such as pectus excavatum, thorax asymmetry, or alterations to the ribs (*p* = 0.083). Similarly, there was no association between the type of surgery and the development of scoliosis or kyphosis (*p* = 0.828).

The choice of thoracotomy or thoracoscopy had no discernible effect upon development of thorax abnormalities (*p* = 0.646), column alterations (*p* = 0.285), or long-term clinical symptoms such as recurrent cough, lower respiratory infections, wheezing, or poor exercise tolerance (*p* = 0.95).

### 3.4. Pulmonary Function Tests

We reviewed results of PFTs carried out after surgery in 54 of 77 children (70.1%), with a median age of 7 years (IQR 5–10 years).

Ten out of these fifty-four patients had obstructive alteration (18.5%), while fifteen experienced restrictive alteration (27.8%).

The values of the investigated parameters are presented in Table 3 as average Z-scores.

Identified pathological patterns in PFTs were not statistically correlated with patient sex, age at surgery, or the presence of symptoms prior to surgery. In addition, there were no significant correlations between PFT alterations and the development of short- or long-term complications after surgery. However, PFTs did reveal a significant correlation between pneumonectomy and the development of lung function deficit, as expressed in a FEV1 z-score of <−2 SD (*p* = 0.031).

## 4. Discussion

CLMs are a heterogeneous group of developmental disorders that affect pulmonary parenchyma, arterial supply, and venous drainage [19]. CLMs are almost always diagnosed as cystic lung lesions by means of a prenatal ultrasound (US). However, their prevalence may be underestimated, as an unknown proportion of these lesions is accidentally diagnosed postnatally [20]. The reported data from local registries and the European Surveillance of Congenital Anomalies (EUROCAT) suggest a higher prevalence than previously supposed [21,22].

Antenatally, suspected CLMs are closely monitored with a serial fetal US to assess their dimensions, localization, and characteristics and for possible signs of fetal hydrops. A fetal MRI may also be used to confirm the presence of a systemic arterial blood supply, to rule out the presence of complications, and to predict the risk of neonatal respiratory distress [23].

However, CLMs can also be diagnosed later in childhood, among children presenting with respiratory symptoms, which need subsequent evaluation by pneumological and imaging methods. In this study, we reviewed the long-term data concerning the clinical outcomes and pulmonary functions of patients with CLMs who underwent pulmonary resection. CLMs include different pathological entities that are distinguished based on their morphological characteristics by postnatal thoracic MRI and CT scans. Kunisaki et al. [2] studied 25 patients with a prenatally diagnosed lung mass who underwent surgical resection. They found that 56% of patients had CPAM, 12% had congenital lobar emphysemas, and 8% had PS, while 24% had CLMs that involved both cystic adenomatoid malformation and bronchopulmonary sequestrations. Among our population, the most frequent diagnosis was CPAM (50.65%), as reported in other studies [24,25], followed by hybrid lesions (25.97%), lobar emphysema (11.69%), and PS (5.19%).

CT scanning is not a reliable method for distinguishing CLM different types. In our study, the pre-resection diagnostic hypothesis was confirmed by a later histological examination in fewer than 50% of patients, without a strong agreement.

These data support the idea that elective surgery might be a better means of differentiating benign forms from those that are malignant, such as pleuropulmonary blastoma. Using CT scanning, it is difficult to distinguish cystic pleuropulmonary blastoma from CAM or CPAM; however, clinical presentation, often spontaneous pneumothorax in infants with negative prenatal imaging, can help in differential diagnosis. [26].

Surgical resection is the appropriate treatment choice for symptomatic patients with CLMs, whether they are diagnosed during fetal life or in childhood. The aim of surgical treatment is to resect the lung tissue alteration in order to alleviate symptoms, improve pulmonary functionality in the remaining lung, prevent long-term complications, and enable compensatory lung growth.

However, the use of resection in asymptomatic CLM cases remains a controversial issue because the surgical risks must be weighed against the potential development of long-term lung and thorax complications.

In our study population, most patients did not present short-term complications after surgery (69%). The presence of short-term complications was not associated with the time or type of surgery (type of approach and extension of the resection) or with the presence of symptoms before surgery. These findings contrast with the results obtained in the meta-analysis by Stanton [27], which highlighted a twofold increase in operative risk in symptomatic patients.

Some authors recommend that elective resections should be performed prior to 10 months of age. In our population, the patients with symptoms were younger, with a lower median age at time of surgery (5 months), which was in line with the findings of other authors that suggest an age range of 3–9 months as the optimal time for intervention to reduce the risk of the development of respiratory symptoms [4,28,29]. As most of our patients underwent surgery in the first year of life, we were unable to highlight any differences in the outcomes relating to time of surgery; the optimal timing for surgery remains a matter of debate.

There is no consistent evidence about the possible effects on long-term pulmonary function [14] resulting from different types and times of surgery; the literature data are contradictory in this regard [30,31].

In our population, we identified a high prevalence of complications during post-surgery follow-up. Clinical or radiological sequelae appeared in 73.7% of patients, which is in line with the 78% of patients who experienced long-term complications in the study by Calzolari et al. [10] that reviewed the case histories of 68 surgically treated CLM children and identified complications including radiological changes, wheezing, lower respiratory tract infections, and chest wall anomalies; we found similar proportions in our population.

There are several hypotheses that can explain the high prevalence of post-surgery sequelae. On the one hand, surgical resection can cause anatomical alterations in the structure of the airways; on the other hand, a malformation may affect a larger part of the respiratory tree, resulting in a greater predisposition to inflammation and bronchial reactivity [32].

However, among our population, only 16 patients exhibited an abnormal lung function. In patients with CLMs, the PFT alterations can be due to the reduction in respiratory reserve, which is proportional to the extent of the anomaly or the amount of resected lung parenchyma [12].

Children’s lungs are capable of organizing regrowth with adequate alveolarization [33]. The catch-up growth of alveolarization extends beyond early childhood into adulthood; however, in humans, it is only in the first periods of life when the generation of new alveoli is sufficiently active to reconstitute an alveolocapillary surface close to the genetic project [33,34,35]. When the growth of the alveolar surface and the growth of the airways is out of phase (disynaptic growth), the compensatory rearrangement of the lung produces a distortion in airway geometry. This leads to an increase in airway resistance, which probably reaches the maximum in children of preschool age [9,36]. This regrowth organization could explain our finding that most of the surgically treated children in our sample entered adult life with normal PFT.

We did not find any correlation between lung function and age at surgery, probably due to the standardized approach, whereby most patients undergo surgery early in life. Our data are similar to those of Keijzer et al. [29], who found that age at lobectomy did not influence FVC or FEV1, when latterly assessed at a mean age of 10 years.

In our population, we found only one significant correlation: between the presence of a lung function deficit (FEV1 Z score < −2 SDs) and the choice of pneumonectomy as the type of surgery used. Due to the highly disabling nature of this surgical approach, this finding was perhaps to be expected.

One rare complication after pneumonectomy that should be considered is post-pneumonectomy syndrome, which is caused by an excessive mediastinal shift. After right-sided pneumonectomy, the left main bronchus may be stretched over the descending aorta, and the lower lobe bronchus is kinked over the descending aorta. The mechanism of functional impairment after left-sided pneumonectomy is less clear, except in patients with a right aortic arch. We found this condition in one patient in our cohort, and this complication is found more frequently in children [37]. Symptoms may include progressive dyspnea on exertion and stridor or tracheomalacia.

There is continuing debate in the literature concerning lung function outcomes in patients surgically treated for CLM. However, studies by Dunn et al. and Lau et al. both suggest that lobectomy for CPAMs has no detrimental effect on lung function in children, and this is in line with our results [38,39]. In addition, the choice of surgical approach—thoracotomy or thoracoscopy—did not lead to differences in terms of early complications, as previously reported [40], and we found no association with lung function outcome.

We acknowledge some limitations in this study. The retrospective design has numerous disadvantages associated with an inferior level of evidence, which limits the validity of our results. We considered minor clinical parameters (e.g., cough, lower respiratory tract infections, and wheezing) to define long-term complication outcomes. This was a consequence of the retrospective nature of our study, which was based on an analysis of patients’ clinical records. The second main limitation of our study was the limited sample size; a larger cohort of patients would be useful for a more significant data analysis. A multicenter, prospective, and randomized study would also help to evaluate data comparing the effects of conservative and elective resection approaches, in terms of clinical and functional outcomes. Finally, we note that no inflammatory biomarkers on histologic examination were considered in our patient cohort; further studies including these parameters could help to define neoplastic risk in asymptomatic patients and give support to the surgical, rather than the conservative, approach [4,32].

## 5. Conclusions

Clinical and functional respiratory complications may occur in children with CLMs undergoing surgical resection. The relationship between short-term clinical complications, young age and early symptoms prior to intervention may support a protective role for elective surgery. During long-term follow-up, clinical outcomes are not related to age, type of surgery, or pre-surgical symptoms. Lung function also needs long-term monitoring, even if it seems to be affected only by the most disabling method of surgical intervention. Finally, a long-term clinical and instrumental follow-up study is needed to improve patient care and to improve the management of asymptomatic patients.

## Figures and Tables

**Table 1 children-09-01881-t001:** Short-term post-surgery complications.

Acute Post-Surgery Complications	No. (24/77)	% (31.2)
Pneumothorax	11/24	45.8
Respiratory distress	4/24	16.7
Hemothorax	1/24	4.2
Bleeding	1/24	4.2
Transient phrenic nerve palsy	1/24	4.2
Anemia	3/24	12.5
Pleural effusion	3/24	12.5
Air-leak syndrome	2/24	8.3
Increased transaminases	1/24	4.2
Pulmonary hypo-inflation	1/24	4.2
Myocardial infarction	1/24	4.2
Femoral deep vein thrombosis	1/24	4.2

No. = number.

**Table 2 children-09-01881-t002:** Long-term post-surgery complications.

	No.	%
**Patients with long-term complications ***	56/76	73.7
Recurrent cough	36/76	44.7
>3 lower respiratory tract infections/year	16/76	21.0
Wheezing	19/76	25.0
Poor exercise tolerance	8/76	10.5
**Altered curvature of the spine ***	9/75	12.0
Scoliosis	8/75	10.7
Kyphosis	1/75	1.3
**Thorax abnormalities**	12/77	15.6
Pectus excavatum	6/77	7.8
Thorax asymmetry	4/77	5.2
Ribs alterations	3/77	3.9
Post-pneumonectomy syndrome	1/77	1.3

* For statistical purposes, only patients with available data were included. No. = number.

**Table 3 children-09-01881-t003:** Pulmonary function tests (PFTs).

PFT (No. 54/77)	Average Z-Score	Standard Deviation
FVC	−1.78	2.24
FEV1	−1.66	2.13
FEV1/FVC	−0.30	1.79
TLC	−0.97	2.40
RV	2.24	2.33
RV/TLC	3.9	3.5
RAW	2.27	2.24

No.—number; FVC—forced vital capacity; FEV1—forced expiratory volume in the 1st second; TLC—total lung capacity; RV—residual volume; RAW—airway resistance.

## Data Availability

The data presented in this study are available on request from the corresponding author.

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
