# Peer review of "Congenital Lung Malformations: Clinical and Functional Respiratory Outcomes after Surgery"

_children, 2022, doi:10.3390/children9121881_

Round 1

Reviewer 1 Report

This study reports on a retrospective study in an extensive cohort of children undergoing surgical resection of LCMs, which contributes to the debate regarding the best treatment strategy.

I have some concerns regarding the manuscript.

One major concern is the language. This manuscript needs  thorough editing. It is full of spelling, language and grammatical issues which makes it difficult to read.

Secondly, the methodology section needs to be reviewed. Why have the different end points been chosen? Why do you use the specific cut oof values. E.g. 3 months of cough: is this cumulative or ongoing cough and how does this relates to a normal pattern of cough in a toddler experiencing multiple viral airway infections. Is this a sensitive and specific outcome parameter. The same counts for respiratory infections: do you mean lower airway infections such as pneumonia treated with antibiotics for example or alle airway infections (including common cold) and do you mean upper airway or lower airway infections or a combination of both? Are these parent reported oucome parameters or doctor diagnosed? If parent reported, have you used a standard questionnaire. If no standard questionnaire is used or the diagnosis is not made by a medical practioner, I have some major concerns with using these parameters as significant outcome parameters. 

With regards to the methodology, this needs to be elaborated more upon in the discussion section.

I would like to see some statistical analyses and values on the association of the surgical technique or the extent of surgery with the different types of complications. It would also be interesting to compare thoracotomy and thorascopy.

In conclusion. I have som major issues regarding language and methodology of this paper and I think the manuscript would benefit from some major editing.

Author Response

This study reports on a retrospective study in an extensive cohort of children undergoing surgical resection of LCMs, which contributes to the debate regarding the best treatment strategy.

I have some concerns regarding the manuscript.

One major concern is the language. This manuscript needs thorough editing. It is full of spelling, language and grammatical issues which makes it difficult to read.

We appreciate your comment and we revised the language.

Secondly, the methodology section needs to be reviewed. Why have the different end points been chosen?

Thank you for this observation. We chose the presence of early or late complications as endpoints of our study. We conducted a retrospective analysis of clinical records, collecting the different clinical complications occurring during the hospitalization for surgical resection and pneumological follow-up. The retrospective nature of data is one of the main limitations of our study.

Why do you use the specific cut oof values. E.g. 3 months of cough: is this cumulative or ongoing cough and how does this relates to a normal pattern of cough in a toddler experiencing multiple viral airway infections. Is this a sensitive and specific outcome parameter.

We chose the definition of persistent cough more than 2 months, as suggested by literature. Cough is a clinical parameter and it has to be considered with other clinical outcomes defining long-term complications.

We specify more deeply this point at lines 96-97.

The same counts for respiratory infections: do you mean lower airway infections such as pneumonia treated with antibiotics for example or alle airway infections (including common cold) and do you mean upper airway or lower airway infections or a combination of both? Are these parent reported oucome parameters or doctor diagnosed? If parent reported, have you used a standard questionnaire. If no standard questionnaire is used or the diagnosis is not made by a medical practioner, I have some major concerns with using these parameters as significant outcome parameters. 

We considered lower airway respiratory infections diagnosed by a doctor (lines 98-99).

With regards to the methodology, this needs to be elaborated more upon in the discussion section.

Thank you for the comment, we update and elaborated on the methodology and discussion sections.

I would like to see some statistical analyses and values on the association of the surgical technique or the extent of surgery with the different types of complications. It would also be interesting to compare thoracotomy and thorascopy.

Thank you for your comment. We elaborated on this analysis in the results section (lines 171-173; lines 187-195).

In conclusion. I have som major issues regarding language and methodology of this paper and I think the manuscript would benefit from some major editing.

Thank you for your comments, we revised the manuscript following your suggestions (please consider the tracked version of revised manuscript).

Reviewer 2 Report

This study investigated the long-term outcomes in children with lung congenital malformations undergoing surgical resection. Overall, the article is comprehensive and the topic is interesting. Yet, I would like to point out the following comments.

Minor comments

Line 21: “The aim of the study was to evaluate...” instead of “The aim of the study was evaluated....”

Major comments

1. Why didn’t the authors include preoperative PFTs in their analysis of data? This would further highlight the effect of surgery on pulmonary function.     

Author Response

This study investigated the long-term outcomes in children with lung congenital malformations undergoing surgical resection. Overall, the article is comprehensive and the topic is interesting. Yet, I would like to point out the following comments.

Minor comments

Line 21: “The aim of the study was to evaluate...” instead of “The aim of the study was evaluated....”

Thank you for the comment, we corrected the sentence in line 21. 

Major comments

  1. Why didn’t the authors include preoperative PFTs in their analysis of data? This would further highlight the effect of surgery on pulmonary function.

Thank you for the comment. We thought that preoperative PFTs could be useful, but we performed a retrospective analysis of our cohort and not all patients undergone preoperative PFTs. So, we can’t include them in the analysis for the lack of data and this is a limit of the retrospective design of our study.  We added this limit in the discussion.

Round 2

Reviewer 1 Report

Thank you for adjusting the manuscript. However, I would really recommend to have it edited by a native English speaker as the adjustments made regarding English language are still insufficient.

Author Response

Thank you very much for your suggestions.

We submitted the manuscript for English editing and we uploaded the revised version.

Reviewer 2 Report

I am satisfied with the authors' responses to my questions/issues raised in my initial review.

Author Response

(The authors gave the same response as above.)
